Metabarcoding analysis reveals hidden eukaryotic plankton biodiversity in the Ross Sea, Antarctica

Choi Soyun 1
Choi Eunkyung 1
Cho Minjoo 1
Lee Seung Jae 1
Kim Inseo 1
Shin Doyoon 1
Kim Jangyeon 1
La Hyoung Sul 2
Rhee Jae-Sung 3
Kim Jeong-Hoon jhkim94@kopri.re.kr 4
Park Hyun hpark@korea.ac.kr 1
1 Department of Biotechnology, College of Life Sciences and Biotechnology, Korea University , Seoul , Republic of Korea
2 Division of Ocean & Atmosphere Sciences, Korea Polar Research Institute (KOPRI) , Incheon , Republic of Korea
3 Department of Marine Science, College of Natural Sciences, Incheon National University , Incheon , Republic of Korea
4 Division of Life Sciences, Korea Polar Research Institute (KOPRI) , Incheon , Republic of Korea
Sun Dong
Electronic publication date: 2025 Oct 14
Publication date: 2025
Volume: 13
Electronic Location ID: e20118
Received 2025 May 27; Accepted 2025 Sep 1
Copyright: ©2025 Choi et al.
Copyright year: 2025
Copyright holder: Choi et al.
License: This is an open access article distributed under the terms of the Creative Commons Attribution License, which permits unrestricted use, distribution, reproduction and adaptation in any medium and for any purpose provided that it is properly attributed. For attribution, the original author(s), title, publication source (PeerJ) and either DOI or URL of the article must be cited.
License URL: https://creativecommons.org/licenses/by/4.0/

Keywords: Ross Sea, Marine protected area, Environmental DNA, Metabarcoding, Eukaryotic diversity, Phytoplankton, Zooplankton

Funding: Korea Institute of Marine Science & Technology Promotion (KIMST) grant funded by the Ministry of Oceans and Fisheries KIMST RS-2022-KS221661 This work was supported by Korea Institute of Marine Science & Technology Promotion (KIMST) grant funded by the Ministry of Oceans and Fisheries (KIMST RS-2022-KS221661) and Korea University grant. The funders had no role in study design, data collection and analysis, decision to publish, or preparation of the manuscript.

==============================
Background

Environmental DNA (eDNA) analysis is a highly sensitive, non-destructive method that enables the detection of various species through DNA shed into environmental samples without requiring direct organism collection. This study sought to investigate the biodiversity and community structure of eukaryotic plankton, including phytoplankton and zooplankton, in the Ross Sea marine protected area (RSR MPA) using eDNA metabarcoding analysis. By examining their spatial and vertical distributions, the study underscores the importance of continuous monitoring for the conservation of the RSR MPA.

Methods

We collected 48 seawater samples from 16 sites in the Ross Sea region for eDNA metabarcoding analysis, targeting the 18S rRNA gene region of eukaryotic plankton in Antarctica. Bioinformatic processing and taxonomic classification were conducted to assess the diversity and community composition of phytoplankton and zooplankton.

Results

Phytoplankton communities were primarily composed of six phyla with their distribution patterns and the grouping of samples with similar community structures was found to be shaped by the ocean currents of the RSR MPA and various environmental factors, such as salinity and dissolved oxygen levels. Zooplankton communities consisted of 18 major taxonomic groups, exhibiting distinct horizontal and vertical distribution patterns with differences in taxonomic community structure and species diversity across depth groups. Notably, previously undetected Antarctic species were identified in the Ross Sea region, demonstrating the effectiveness of eDNA in revealing hidden biodiversity.

Conclusions

Analyzing eukaryotic plankton communities in the vast and extreme Antarctic environment based on eDNA has proven to be highly efficient, enabling the detection of a greater number of species, including those that were difficult to identify in previous studies. It was observed that in the Ross Sea Marine Protected Area, various species form distinct community structures such as phytoplankton and zooplankton, each inhabiting the area according to different environmental variables and habitat preferences. As a designated marine protected area, the Ross Sea’s unique ecosystem requires continuous monitoring and conservation efforts to address environmental changes. The genetic data obtained in this study contributes to expanding the database of Antarctic-specific species, facilitating more accurate and efficient analyses of Antarctic ecosystems in the future.

Introduction

The Ross Sea is a deep embayment located in the southern Antarctic region, adjacent to the Ross Ice Shelf. This region harbors remarkably high biodiversity among Antarctic marine areas, making it a valuable site for studying ecological communities to support conservation and effective management (Smith Jr et al., 2012). The Commission for the Conservation of Antarctic Marine Living Resources (CCAMLR) designated the Ross Sea as a marine protected area (MPA) in 2016 (under CCAMLR Conservation Measure 91-05) (Brooks et al., 2020). Covering nearly 1.55 million km2, the Ross Sea is the world’s largest MPA (Giorli & Pinkerton, 2019). To understand the overall structure of the ecosystem in this extensive Ross Sea region MPA (RSR MPA), it is necessary to identify the present species and assess their diversity.

However, Antarctica’s geographic isolation and extreme cold climate make access difficult (Peck, 2018; Vyverman et al., 2010). This region is also highly vulnerable to anthropogenic activities, particularly in the Ross Sea, where excessive fishing and large-scale sampling of inhabiting species pose risks to the environmental stability of the MPA (Pertierra et al., 2021; Postigo et al., 2023). Furthermore, studying organisms in deep marine environments, such as benthic zones, is uniquely challenging due to the limitations of direct sampling and species observation (Gage & Bett, 2005; Laroche et al., 2020). Given these challenges, research approaches that enable accurate and cost-efficient biodiversity assessments, beyond traditional hunting and sampling method, are essential (Beng & Corlett, 2020; Thomsen & Willerslev, 2015).

Environmental DNA (eDNA) analysis offers an effective alternative. eDNA consists of a complex mixture of DNA molecules shed by organisms within a given environment (Taberlet et al., 2012). It enables the identification of various species indirectly through environmental samples, such as seawater, freshwater, and soil, without the need for direct organism collection (Deiner et al., 2015; Ruppert, Kline & Rahman, 2019). eDNA analysis is a highly sensitive approach for species detection, making it particularly useful for identifying organisms that are difficult to classify visually or capture directly (Ficetola et al., 2008; Smart et al., 2015). eDNA metabarcoding allows for the simultaneous detection of multiple species by targeting conserved genetic sequences shared among specific taxa (Ruppert, Kline & Rahman, 2019). In addition to metabarcoding, eDNA can also be used to detect specific organisms, such as invasive or rare species (Larson et al., 2020; Xia et al., 2021). Unlike conventional organism collection methods, eDNA analysis is non-destructive, facilitates safer sampling, and enables the rapid and accurate identification of diverse organisms from environmental samples (Coble et al., 2019; Van Der Heyde et al., 2020). These advantages make eDNA an ideal tool for studying ecological community structures in vast and extreme environments like Antarctica (Howell, LaRue & Flanagan, 2021). In recent years, eDNA-based study in the Antarctic ecosystem has expanded, focusing on terrestrial plants, animals, microorganisms, and marine organisms (Carvalho-Silva et al., 2021; Cowart, Murphy & Cheng, 2018; Howell, 2021; Zhang et al., 2024).

The conservation of Antarctic organisms and ecosystems has recently garnered increasing attention, driven by concerns over rapid climate change and increasing anthropogenic activities (Chown et al., 2012; Senatore, 2023). Antarctic endemic species are distinct from organisms in other regions and hold significant research value (Vyverman et al., 2010). Having evolved in geographic isolation and extreme cold, Antarctic biota have developed unique adaptation mechanisms to survive in such harsh conditions (Daane & Detrich III, 2022; Peck, 2018; Rogers, 2007). Therefore, studying these extremophiles provides insights into how organisms adapt to environmental stresses and develop survival strategies, while also offering critical biological and ecological knowledge due to their high sensitivity to environmental changes (Eason et al., 2016; Pearce, 2012). The Ross Sea region hosts a diverse array of organisms, including microorganisms, plankton, and eukaryotes such as krill, as well as marine fauna such as fish, penguins, seals, and whales (Davis et al., 2017; Hanchet et al., 2013; Li, Gu & Gui, 2020; Nie et al., 2012; Smart et al., 2015; Van Dam & Kooyman, 2004). While various studies have been conducted on the Ross Sea ecosystem, eDNA analyses in the RSR MPA have been relatively limited, often focusing on specific regions or taxa (Jeunen et al., 2024; Jeunen et al., 2023). Additionally, large-scale seawater sampling studies have not yet been conducted. To accurately represent the biodiversity of the vast Ross Sea region, comprehensive eDNA analyses covering a wide range of locations and depths are essential.

This study aims to investigate the overall biodiversity of eukaryotic plankton in the RSR MPA using marine eDNA analysis. Seawater samples were collected across a horizontally extensive area to represent the broader Ross Sea ecosystem, and samples were also taken at various depths to examine vertical distribution patterns. Phytoplankton, as primary producers, play a crucial role in Antarctic oceanic photosynthesis, while zooplankton, are key components in maintaining the balance of the Antarctic ecosystem and its complex food web (Hernando et al., 2015; Jia et al., 2016). By analyzing eukaryotic plankton communities, this study aims to identify major community structures and habitat distributions within the RSR MPA, providing valuable insights for the conservation and functional preservation of the Antarctic ecosystem.

Materials & Methods

Seawater sampling and eDNA extraction

Seawater sampling was conducted in the Ross Sea, Antarctica, using Niskin bottles attached to a Conductivity-Temperature-Depth (CTD) rosette sampler abroad the Korean icebreaking research vessel (IBRV) Araon from January 21 to February 1, 2023. A total of 48 seawater samples were collected from 16 sites, with three different depths per site. Seawater samples were grouped into the following depth ranges; Surface (0 m), Epipelagic (0–200 m), Mesopelagic (200–1,000 m), and Bathypelagic (>1,000 m), with each site represented in three ranges (Fig. 1, Table 1). All samples were frozen at −20 °C until filtration and DNA extraction.

Figure 1 Sampling sites in the Ross Sea, an Antarctic Marine Protected Area.

A total of 48 seawater samples were collected from various depths across 16 sites and analyzed using eDNA metabarcoding targeting the 18S rRNA region to assess the biodiversity of eukaryotic organisms, including phytoplankton and zooplankton. The white line on the map represents the boundary of the MPA.

Table 1 Sampling site information of Antarctic seawater samples.

Seawater samples from the Ross Sea region were categorized by depth range: Surface (S; 0 m), Epipelagic (E; 0–200 m), Mesopelagic (M; 200–1,000 m), and Bathypelagic (B; >1,000 m).

Longitude	Latitude	Station	Depth (m)	Group		Longitude	Latitude	Station	Depth (m)	Group	
			271	M					300	M	
164.10 E	74.64 S	12	100	E		180.00 E	73.80 S	46	150	E	
			0	S					0	S	
			491	M					400	M	
163.83 E	74.92 S	16	200	E		180.00 E	74.90 S	50	120	E	
			0	S					0	S	
			1000	B					330	M	
165.47 E	75.03 S	18	200	E		180.00 E	76.10 S	52	60	E	
			0	S					0	S	
			300	M					2000	B	
170.70 E	75.91 S	26	100	E		175.00 W	76.10 S	67	40	E	
			0	S					0	S	
			573	M					2638	B	
171.00 E	76.10 S	29	150	E		175.00 W	73.80 S	69	200	E	
			0	S					0	S	
			600	M					2596	B	
175.50 E	77.40 S	35	180	E		170.00 W	74.90 S	71	200	E	
			0	S					0	S	
			500	M					565	M	
175.50 E	76.10 S	38	100	E		170.00 W	76.10 S	77	200	E	
			0	S					0	S	
			570	M					532	M	
175.50 E	74.80 S	44	200	E		170.00 E	77.80 S	80	200	E	
			0	S					0	S	

Each 2-L seawater sample was vacuum-filtered using a cellulose acetate (CA) filter with a 0.22 µm pore size (Corning). eDNA extraction was performed on a quarter piece of the filter using the FastDNA™ SPIN Kit for Soil (MP Biomedicals), and the extracted DNA was eluted in 100 µL of DNase-Free Water (DES). The concentration and quality of the extracted DNA were assessed using a Nanophotometer® NP80 spectrophotometer (Implen).

Amplification of eukaryote 18S rRNA and sequencing

The V4 hypervariable region of the eukaryotic 18S rRNA gene was amplified via polymerase chain reaction (PCR) using the TAReuk454FWD1 (5′- CCA GCA SCY GCG GTA ATT CC-3′) and TAReukREV3 (5′-ACT TTC GTT CTT GAT YRA-3′), universal primers as described previously (Liu et al., 2021), which is verified from the raw origin of this primer set (Stoeck et al., 2010). The PCR reaction mixture contained 25 µL of 2X EmeraldAmp Max PCR Master Mix (Takara Bio), one µL of each primer (20 pmol), 100 ng of template DNA, and distilled water to final volume of 50 µL. The thermal cycling conditions consisted of initial denaturation at 98 °C for 2 min, followed by 25 cycles of 98 °C for 10 s, 53 °C (for the first 10 cycles), 48 °C (for the remaining 15 cycles) for 30 s, and 72 °C for 30 s, with a final extension at 72 °C for 2 min. The size of the first PCR products (∼444 bp) was verified using 1.5% agarose gel electrophoresis and purified using the LaboPass™ Gel and PCR Clean-up Kit (Cosmo Genetech). A second PCR was performed using the same primers, tagged with adapter and index sequences for library construction. The reaction mixture for the second PCR contained 25 µL of 2X EmeraldAmp Max PCR Master Mix, one µL of each primer (20 pmol), four µL of purified first PCR product, and distilled water up to 50 µL. The thermal cycling conditions were as follows: initial denaturation at 98 °C for 2 min, followed by 30 cycles of 98 °C for 10 s, 68 °C for 30 s, and 72 °C for 30 s, with a final extension at 72 °C for 2 min. After purifying the products of the second PCR, quantification was performed using the qPCR Quantification Protocol Guide (KAPA Library Quantification Kits for Illumina Sequencing Platforms) and assessed using the LabChip GX HT DNA High Sensitivity Kit (Perkin Elmer). Paired-end (2 × 300 bp) sequencing was conducted on the Illumina MiSeq™ platform.

Bioinformatic data analysis

Raw sequence data obtained from the Illumina MiSeq platform were imported as demultiplexed reads into Quantitative Insights into Microbial Ecology 2 (QIIME2 2023.2; https://qiime2.org/) (Bolyen et al., 2019). Non-biological sequences, such as adapters and primers, were trimmed using the cutadapt plugin (Martin, 2011). Denoising and clustering were performed using the vsearch plugin (Rognes et al., 2016). After which denoised sequence reads were clustered into operational taxonomic units (OTUs) at 97% sequence similarity. Chimeric sequences were removed from the OTUs through de novo detection using the uchime-denovo algorithm (Edgar et al., 2011). Chloroplast and mitochondrial sequences were also excluded. For taxonomic classification, query sequences were compared to the SILVA reference database (https://www.arb-silva.de/), followed by taxonomy alignment (Quast et al., 2012).

The assigned eukaryotic organisms were broadly divided into two major groups for analysis: phytoplankton, which perform photosynthesis and serve as primary producers in the marine ecosystem, and zooplankton, which are heterotrophic plankton that consume other organic matter and include a variety of animal taxa in their larval forms (Conway, 2012; Mwagona, Chengxue & Hongxian, 2018; Sildever et al., 2021). Surface water samples were analyzed for phytoplankton analysis, while all samples were used for zooplankton analysis. The taxonomic composition of marine eukaryotic plankton communities was assessed from the kingdom to species level. Alpha and beta diversity analyses were conducted separately for phytoplankton and zooplankton classification results, using classifiers customized based on OTUs assigned to each group. Alpha diversity of eukaryotic plankton in Antarctic seawater samples was evaluated using Observed Features, Chao1, Faith’s Phylogenetic Diversity (Faith PD), and Abundance-based Coverage Estimation (ACE) for species richness, along with Simpson, Shannon-Wiener, and Pielou’s Evenness indices for species diversity and evenness. Beta diversity analysis for phytoplankton and zooplankton communities was performed using principal coordinate analysis (PCoA) based on Bray-Curtis (presence and abundance-based) and Weighted UniFrac (abundance and phylogenetic tree based) distances, respectively (Baselga, 2013; Chang, Luan & Sun, 2011). The grouping of phytoplankton samples within the unweighted pair group method with arithmetic mean (UPGMA) tree was based on the Jaccard similarity distance, which as a presence/absence-based index, clearly illustrates overall compositional turnover and enables grouping of samples solely according to shared taxa without distortion from abundance differences (Mainali et al., 2017). Data visualization including UPGMA tree, OTU Venn diagram and heatmap were conducted using R (v4.3.2) within the RStudio (v2023.12.0-369) environment.

The phytoplankton and zooplankton species identified in this study are listed in Table 2. We compared these results with data from the Scientific Committee on Antarctic Research (SCAR, https://www.biodiversity.aq/) and the Register of Antarctic Marine Species (RAMS, https://www.marinespecies.org/rams/index.php) (De Broyer et al., 2023), as well as previous studies, to determine whether these species have been previously recorded in Antarctica.

Table 2 Comparison of species-level assigned taxonomy from this study with previous records in Antarctica.

“Observed” indicates whether the species has been directly observed in the Antarctic region, while “eDNA” represents detection through environmental DNA analysis from samples such as seawater. The “Ross Sea” column specifies whether the species has been recorded in the Ross Sea region. “+” denotes presence, a “–” indicates absence, and “+-” indicates detection applicable to organisms of the same genus.

Assigned taxonomy (phytoplankton)	Observed	eDNA	Ross Sea		Assigned taxonomy (zooplankton)	Observed	eDNA	Ross Sea	
Actinocyclus actinochilus	+	+	–		Acanthocystis haeckeli	–	+	+	
Biecheleria brevisulcata	–	+	+		Amphibelone anomala	+	+	+	
Chaetoceros brevis	+	+	+		Amphicorina ascidicola	+-	–	+-	
Chaetoceros dichaeta	+	+	+		Amphilonche elongata	–	+	–	
Chaetoceros neogracilis	+	+	+		Brada villosa	+	–	–	
Corethron inerme	+	+	+		Calliacantha natans	+	+	+	
Cylindrotheca closterium	+	+	+		Cephalothrix rufifrons	–	+-	–	
Dinovorax pyriformis	–	+	–		Cyclotrichium cyclokaryon	+-	+	–	
Eucampia antarctica	+	+	+		Edwardsiella lineata	+-	+-	+-	
Fragilariopsis kerguelensis	+	+	+		Gromia winnetoui	+	+	–	
Geminigera cryophila	+	+	+		Halomonhystera disjuncta	+	+-	+	
Hemistasia phaeocysticola	–	+	+		Labyrinthulid quahog	+-	+	–	
Impagidinium pallidum	+	+	–		Lankesteria cystodytae	–	+	–	
Karlodinium veneficum	+-	+	+		Lepidonotus sublevis	+-	+	–	
Lessardia elongata	+-	+	–		Parvamoeba rugata	–	+	–	
Mantoniella antarctica	+	+	+		Parvilucifera prorocentri	–	+	+	
Margalefidinium fulvescens	+-	+	+		Pentapharsodinium tyrrhenicum	+-	+	+	
Monostroma grevillei	+	+	+		Pirum gemmata	–	+	–	
Phaeocystis antarctica	+	+	+		Pista cristata	+	+-	+	
Porosira pseudodenticulata	+	+	+		Pseudotrachelocerca trepida	+	+	–	
Protoperidinium bipes	+	+	+		Salpingoeca oahu	+-	+-	+-	
Protoperidinium depressum	+	+	+		Spumellaria radiolarian	+-	+	+	
Pterosperma cristatum	+	+	+		Thouarella antarctica	+	+	+	
Pyramimonas australis	+	+	+						
Pyramimonas gelidicola	+	+	+						
Rhizosolenia fallax	+-	+	–						
Thalassiosira tumida	+	+	+						

Results

Sampling and sequencing results

A total of 48 seawater samples were collected from the Ross Sea, Antarctica. Environmental parameters, including temperature, salinity, and dissolved oxygen, were measured during sampling using a CTD. The values for each sample are summarized in Table S1. The highest seawater temperature (0.85 °C) was recorded in a surface sample from Station 12, while the lowest (−2.04 °C) was observed in an epipelagic sample from Station 35. The average temperature across all samples was −1.13 °C. Salinity ranged from 33.56 PSU in a surface sample from Station 26 to 34.89 PSU in a bathypelagic sample from Station 18, with an average salinity of 34.41 PSU. Dissolved oxygen (DO) levels were lowest in a bathypelagic sample from Station 67 and highest in a surface sample from Station 16, ranging from 227.29 to 404.21 µmol/kg, with an average of 318.95 µmol/kg.

Sequencing using the Illumina MiSeq platform yielded a total of 71,582 OTUs and 4,833,418 reads from the 48 Antarctic seawater samples. Among total OTUs and reads, the proportion classified for ‘phytoplankton’ is 41,277 OTUs and 3,554,149 reads, while for ‘zooplankton’ it is 12,762 OTUs and 974,893 reads. The number of OTUs and reads for each sample is presented in Table S2. Taxonomic classification revealed that 76.45% of OTUs (54,723) and 94.57% of reads (4,570,796) were assigned at the phylum level. At the species level, 13.05% of OTUs (9,338) and 19.26% of reads (930,950) were successfully classified (Table S3).

Phytoplankton diversity in the Ross Sea

Analysis of the taxonomic composition of phytoplankton in surface samples was conducted at both the phylum and genus level (Fig. 2). At the phylum level, the surface samples were mainly represented by the following six phyla: Dinoflagellata, Diatomea, Cryptophyta, Chlorophyta, Prymnesiophyta, and Euglenozoa (Fig. 2A). Dinoflagellata was the most abundant taxon, followed by Diatomea. Notably, Cryptophyta was more abundant at offshore stations such as Stations 46, 50, and 67. At the genus level, 35 phytoplankton genera were predominantly detected in surface samples (Fig. 2B). Gymnodinium and Pseudo-nitzschia, both representative Antarctic plankton, were prevalent across most stations. Specifically, the genus Corethron (Diatomea) was more abundant at offshore stations (38, 44, 46, and 50) and Fragilariopsis (Diatomea) was generally more abundant at the nearshore stations (12, 16, and 18) and at station 71.

Figure 2 Taxonomic composition of phytoplankton in each surface sample at the (A) phylum and (B) genus levels.

Site numbers are indicated at the bottom of each bar in the graph. The surface samples from each site were primarily composed of six phyla and 35 genera of phytoplankton.

A UPGMA tree was constructed to cluster samples with similar species compositions based on the Jaccard similarity distance (permutational analysis of variance (PERMANOVA): F-value 1.63; p-value < 0.001) (Fig. 3). The seawater samples were broadly classified into four groups based on the relative abundance of six major species. The map on the right in Fig. 3 illustrates the grouping results based on ocean currents in the Ross Sea region, alongside the clustering results from the UPGMA tree analysis. Ocean currents in the MPA were represented as gray arrows on the background of the map in Fig. 3, based on results from previous studies (Sedwick et al., 2011; Smith Jr et al., 2014). The grouping of samples with similar taxonomic community structures, as revealed by the UPGMA tree, corresponded to the grouping patterns influenced by ocean currents including the Ross Sea.

Figure 3 UPGMA tree of phytoplankton based on Bray-Curtis distance, combined with species composition and relative abundance of the six major species.

By comparing the grouping results of surface seawater with the flow of ocean currents (gray arrows) in the Ross Sea region, groups with similar taxonomic compositions were found to be influenced by ocean currents, with Groups 3 and 4 being significantly affected by the Ross Sea Gyre.

The diversity of phytoplankton communities in Antarctic seawater was assessed using alpha and beta diversity analyses based on taxonomic assignment results (Fig. 4). Alpha diversity indices were used to measure species richness and evenness within each sample group (Fig. 4A). Among the four groups, Group 1 exhibited the highest species diversity, while Group 4 had the lowest. Group 1 primarily consisted of coastal samples, which generally exhibited higher species richness and evenness. Beta diversity, assessed using PCoA based on the Bray-Curtis distance, showed that samples clustered distinctly into four groups (Fig. 4B). Additionally, these groups were further categorized into coastal region sites (Groups 1 and 2) and open sea region sites (Groups 3 and 4). The OTU Venn diagram illustrates the distribution of OTUs among the four groups (Fig. 4C). A total of 2,148 OTUs were shared across all groups. The highest number of unique OTUs was found in Group 3 (8,093 OTUs), while Group 4 had the lowest (4,523 OTUs). Group 1, which was closest to the coast, had the second-highest number of unique OTUs (6,313 OTUs).

Figure 4 Diversity analysis results for each phytoplankton group.

(A) Alpha diversity and (B) beta diversity (based on Bray-Curtis distance) results of each group. (C) OTU Venn diagram of phytoplankton in each group. G1 to G4 represent Group 1 to Group 4, respectively. Species diversity was highest in Group 1 and lowest in Group 4. Seawater samples were distinctly clustered by group, with further classification into coastal sites (Groups 1 and 2) and open sea sites (Groups 3 and 4). The dots inside the box at the bottom right of (B) represent their size according to the number of OTUs.

Zooplankton diversity in the Ross Sea

The taxonomic composition of zooplankton mainly focused on the phylum level categorized into four depth-based groups (Fig. 5). We primarily focused on the phylum level for taxonomic classification of zooplankton. However, due to the current limitations and structure of reference databases, some groups assigned at the subphylum or clade level, where formal phylum level classification is not yet established or remains under debate, were also included in the phylum level analysis. A total of 18 major taxonomic groups were identified in the zooplankton community. In the Surface group, Ciliophora and Protalveolata were dominant, with Ciliophora particularly abundant in coastal regions such as Stations 12, 16, 18, and 80. Notably, Arthropoda was more prevalent at specific stations, such as 69 and 71, compared to other regions. Picozoa was more abundant in open-sea regions, including Stations 46 and 67, than in coastal areas. In the Epipelagic group, Protalveolata was the most abundant taxon, with Retaria showing an increased proportion compared to the Surface group. Although Ciliophora was dominant in the Surface group, it remained abundant in specific regions of the Epipelagic group, particularly at Station 12 and 52. Interestingly, Ctenophora was dominant in Station 71, likely due to the local aggregation or movement of its population. In the Mesopelagic group, Protalveolata and Retaria remained highly abundant, similar to the Epipelagic group. Retaria was particularly prevalent in coastal regions such as Stations 35 and 80. Picozoa was relatively abundant in offshore regions, including Stations 44, 46, 50 and 52. In the Bathypelagic group, Protalveolata had the highest proportion among all samples, while Arthropoda was notably abundant at Station 18, the only coastal station in the Bathypelagic group. We also analyzed the taxonomic composition of zooplankton at the order level and detected 44 orders, which will be discussed in relation to the phylum-level taxa in the following discussion section.

Figure 5 Taxonomic composition zooplankton at the phylum level.

Each site number is indicated at the bottom of each bar in the graph. Zooplankton samples from each site were primarily composed of 21 phyla.

To provide an intuitive visualization of the relative abundance of major zooplankton phyla, we presented a heatmap based on the number of sequencing reads (Fig. 6). Protalveolata was the most abundant taxon across all samples, displaying an increasing trend in read counts with depth, reaching its highest abundance in the Bathypelagic group. Retaria and Picozoa were more prevalent in the deeper Epipelagic and Mesopelagic groups than in the Surface group, while Ciliophora exhibited the highest read count in the Surface group. Holozoa had relatively low read counts compared to other major phyla and was primarily detected in the Epipelagic and Mesopelagic groups. Arthropoda was mostly found in the Surface and Epipelagic groups, which are relatively shallow, but also exhibited high read counts in the coastal regions of the deeper Bathypelagic group. Ctenophora was observed in large quantities at specific sites within the Epipelagic group.

Figure 6 Heatmap representing the number of reads based on the composition of major zooplankton phyla in each sampling site and depth of the samples.

Color scale indicates the abundance of zooplankton (read counts). Darker red represents higher values, while lighter colors indicate lower values.

The diversity of zooplankton communities in Antarctic seawater was assessed using alpha and beta diversity analyses (Fig. 7). Alpha diversity indices were used to measure species richness and evenness for each depth-based group (Fig. 7A). Among the four groups, the surface group exhibited the highest species richness, while the Bathypelagic group had the highest species evenness, followed by the Surface group. Beta diversity, assessed using PCoA based on the Weighted unique fraction metric (UniFrac) distance, revealed that samples clustered according to seawater depth (Fig. 7B). The Surface and Bathypelagic groups were relatively distinct from each other, whereas the Epipelagic and Mesopelagic groups were clustered more similarly. Figure 7C presents a Venn diagram of the distribution of OTU numbers across the four depth-based groups. A total of 1,926 OTUs were shared across all groups. The highest number of unique OTUs was observed in the Surface group (21,585 OTUs), while the Bathypelagic group had the lowest (6,661 OTUs). As depth increased, the number of unique OTUs showed a decreasing trend. The OTUs consistently shared across all groups are inferred to belong primarily to Protalveolata, as this group was highly abundant across all depth zones.

Figure 7 Diversity analysis results of each zooplankton group.

(A) Alpha diversity and (B) beta diversity (based on the Weighted UniFrac distance) results of each group. (C) OTU Venn diagram of zooplankton in each group. Group names are abbreviated as follows: S (Surface), M (Mesopelagic), E (Epipelagic), and B (Bathypelagic). Species richness was highest in the Bathypelagic group, while the Surface group exhibited the highest species evenness. Each group (Groups 1, 2, 3, and 4) showed a clustering pattern according to depth based on the Principal Coordinate Analysis (PCoA) results. The dots inside the box at the bottom right of (B) represent their size according to the number of OTUs.

Newly identified Antarctic phytoplankton and zooplankton species

In this study, we identified 27 species of phytoplankton and 23 species of zooplankton. Among the 27 phytoplankton species detected, 20 had been previously observed in the Antarctic region, excluding three species that had not been recorded and four species that were only identified at the genus level. However, all 27 species identified in this study had been detected by eDNA metabarcoding analysis in previous studies. In the Ross Sea region, the focal area of this study, all species had been recorded in previous study except for Actinocyclus actinochilus, Dinovorax pyriformis, Impagidinium pallidum, Lessardia elongata, and Rhizosolenia fallax.

Compared to phytoplankton, which abundantly inhabit in surface waters, zooplankton relatively inhabit a wide range of habitats, including both pelagic and benthic areas, making direct observation more challenging. Of the 23 zooplankton species identified, only eight species were directly observed in previous studies. In contrast, previous eDNA metabarcoding analysis successfully detected 11 of the 15 species that were either not visually observed or could not be classified at the species level. Furthermore, among the species identified in this study, 12 species (Amphilonche elongata, Brada villosa, Cephalothrix rufifrons, Cyclotrichium cyclokaryon, Gromia winnetoui, Labyrinthulid quahog, Lankesteria cystodytae, Lepidonotus sublevis, Parvamoeba rugata, Pirum gemmata, and Pseudotrachelocerca trepida) had not been previously recorded in the Ross Sea region. Additionally, three species (Amphicorina ascidicola, Edwardsiella lineata, and Salpingoeca oahu) had only been identified at the genus level in prior studies.

Discussion

In this study, we performed eDNA metabarcoding analysis to investigate the eukaryotic plankton communities of the Ross Sea Marine Protected Area, one of the most biologically productive regions, which plays a crucial role in global carbon cycling and the polar marine ecosystem (Smith Jr et al., 2012). Directly surveying the biota across the vast Ross Sea region presents significant challenges, particularly in deep-sea environments characterized by extreme conditions such as low temperatures and high pressure, which limit accessibility. We applied a method for analyzing seawater eDNA to detect a broad range of taxa within the extreme polar region, while minimizing environmental disturbance.

This study provides a comprehensive analysis of the biodiversity and community structure of eukaryotic plankton in the Ross Sea, contributing to a deeper understanding of the Antarctic marine ecosystem and its responses to environmental variability. To allow for robust amplification across a broader range of eukaryotic plankton and provide a comprehensive overview of community structure, we targeted the 18S rRNA region with a universal primer set in this metabarcoding analysis. Furthermore, many Antarctic species remain genetically underexplored. By utilizing the genetic data obtained in this study, we anticipated that the establishment and expansion of Antarctic-specific genetic databases, such as Scientific Committee on Antarctic Research (SCAR) and Register of Antarctic Marine Species (RAMS), will facilitate more accurate and efficient study on Antarctic biodiversity.

Phytoplankton communities in the Ross Sea influenced by environmental factors

In this study, we detected six major phytoplankton phyla in surface water samples from the Ross Sea. Among them, Dinoflagellata and Diatomea were the most abundant, consistent with their status as two of the most common phytoplankton groups in seawater. Additionally, Cryptophyta, which was detected in high abundance in regions farther from the coast, is known for its dominance in central sea regions during the phytoplankton spring bloom (Gieskes & Kraay, 1983). At the genus level, Gymnodinium, a representative Antarctic dinoflagellate, and Pseudo-nitzschia, a representative Antarctic diatom, were detected in all samples. The genus Corethron was found in high abundance in regions, which exhibited relatively higher salinity levels and located farther from the coastal sea. Corethron species are known to exhibit morphological changes in response to decreasing salinity and, therefore, cannot thrive in coastal regions where salinity fluctuates significantly due to various factors such as glacial melt (Aizdaicher & Markina, 2010). Meanwhile, Fragilariopsis was more abundant at four sites (Stations 12, 16, 18, and 71) with the highest DO levels. Prior study indicates that certain Fragilariopsis species, such as the sea ice diatom, are more prevalent in oxygen-rich regions within the Antarctic marginal ice-edge zone (Kang & Fryxell, 1992). The genus Chaetoceros, belonging to the family Mediophyceae (Diatomea), was most abundant at site 44, followed by sites 26 and 29. Given the ocean currents in the Ross Sea, these three sites are likely influenced by the same current flow. It is plausible that a temporary bloom of Chaetoceros, which thrives in nutrient-rich environments, was triggered by nutrient input from melting icebergs in the surrounding waters (Hoffmann, 2007). UPGMA tree results based on the Jaccard suggest that groups of samples with similar taxonomic compositions were structured according to the ocean currents in the Ross Sea. The Ross Sea Gyre, which results from interactions between the Antarctic continental shelf and the Antarctic Circumpolar Current, appears to have played a major role in forming Group 3 along the clockwise-flowing current. Notably, Group 3 surrounds Group 4, likely contributing to the ecological isolation of Group 4 from other groups. This isolation is further supported by the diversity analysis and OTU Venn diagram results. Overall, phytoplankton communities in the Ross Sea appear to be shaped by ocean currents as well as environmental factors such as temperature, salinity, and DO levels. Since these organisms mainly inhabit the surface region, which is directly exposed to the external environment, they are inevitably more affected by various environmental changes. Consequently, they are expected to be particularly vulnerable to environmental changes in Antarctica, including those driven by climate change and rising sea temperatures. To support the conservation and management of Antarctic phytoplankton, periodic ecosystem monitoring using safe and non-invasive methods, such as eDNA analysis, will be essential.

Vertical and horizontal distribution patterns of zooplankton in the Ross Sea

In this study, we detected 18 main phyla of zooplankton and the taxonomic composition at the order level was also analyzed (Fig. S1). Ciliophora, which is known as a component of protozooplankton and is typically found in coastal regions, was highly abundant at coastal stations in the Surface group (Macek et al., 2022). At the order level, the Conthreep clade was predominantly represented among these coastal species (Choi et al., 2020). Calanoida, belonging to the phylum Arthropoda, was particularly abundant at Station 69 and 71 in the Surface group, where low salinity values were present. This observation is consistent with a previous study, which indicates that Calanoida have a wide range of adaptations to varying salinity levels (Cervetto, Gaudy & Pagano, 1999). The phylum Picozoa was more abundant in open ocean regions than in coastal areas. These species are typically found in nutrient-limited or oligotrophic environments, such as the open sea (De la Peña et al., 2021). Syndiniales, belonging to the sub-phylum Protalveolata, were abundant in both the Surface and Epipelagic groups, showing an increasing trend with depth. Syndiniales are globally distributed parasites, found even in the Antarctic and Arctic poles. They exhibited the highest OTU counts when parasites were analyzed across various depths (from five m to over 1,000 m) of seawater and sediment in Antarctica (Cleary & Durbin, 2016). Retaria typically known to inhabit depths from the surface to 200 m, were more abundant in the Epipelagic group than in the Surface group (Munir, Sun & Morton, 2021). Ctenophora, which made up a large proportion of the zooplankton at Station 71 in the Epipelagic group, was confirmed to belong to the order Cydippida. In the Mesopelagic group, both Protalveolata and Retaria were detected in high abundance, with Nessellaria, a subphylum of Retaria, making up a large proportion at Stations 35 and 80, which are located near the continent. It is known that large deep-sea agglutinated Foraminifera (a subphylum of Retaria) are predominantly found in nutrient-rich environments near continental margins (Gooday, Shires & Jones, 1997). Similar to the Surface group, Picozoa was found abundantly in the Mesopelagic group at stations located far from the coast. In the Bathypelagic group, Syndiniales from the sub-phylum Protalveolata represented the highest proportion and abundance of reads among all groups. Additionally, Arthropoda was specifically abundant at Site 18, which is the only coastal location among the Bathypelagic groups. Most of the organisms within this phylum were further assigned to the class Maxillopoda or the oreder Calanoida, consistent with reports that these taxa typically inhabit littoral and benthic regions during their life cycles (Glippa et al., 2011; Suárez-Morales, 2015). The distribution of the zooplankton community appears to be influenced not only by the horizontal location of the stations but also by vertical depth. This contrasts with phytoplankton, which are limited to surface samples. Diversity analysis indicates clear separation among groups at each depth range, with the Surface and Bathypelagic groups exhibiting opposite patterns.

Current limitations of the study and future perspectives

Through this study, we identified various Antarctic plankton species that had been previously observed or detected via eDNA in Antarctica but were not found in the Ross Sea region. For phytoplankton, most species detected in this eDNA metabarcoding analysis were previously identified through visual surveys, though some remained unclassified at the species level. While surface-inhabiting phytoplankton are relatively accessible for visual confirmation, eDNA analysis enhances the species detection by overcoming observational limitations. For zooplankton, eDNA analysis enabled species-level identification in deep-water zones where direct biological sampling is challenging. Also, for many organisms that could not be identified to the species level through traditional observation methods, clear species-level classification was achieved using eDNA analysis.

Generally, eDNA metabarcoding enables precise species classification without the need for physical specimens and is both cost- and time-effective for broad-scale biodiversity assessments in logistically challenging environments. However, this approach also has inherent limitations, including PCR primer biases, differential DNA recovery rate during extraction, and gene copy number variations across taxa. Also, the compositionality of sequencing data can further complicate absolute quantification and insufficient information on rare taxa leads to incomplete reference databases, which critically limit taxonomic resolution. To overcome these limitations, future eDNA metabarcoding studies should make efforts to establish standardized experimental protocols that can be applied consistently, and ongoing research efforts should focus on supplementing reference databases to enable more accurate analytical results. Furthermore, continuous and extensive further studies are required to collect a wider range of environmental variable data, enabling the assessment of how these factors directly impact the ecological communities in the Ross Sea region.

Given the broad scope of this study, further research is needed to explore how these species interact with the Antarctic environment. The findings from this study are expected to serve as a foundation for future Antarctic biological study and contribute to advancing the understanding of the Antarctic ecosystem.

Conclusions

In this study, we investigated the marine eukaryotic plankton inhabiting the Ross Sea, Antarctica, through eDNA metabarcoding analysis. Our results revealed that the abundance of certain phytoplankton species in specific surface regions was influenced by particular environmental factors. In contrast, considering the ecological characteristics of zooplankton, which inhabit a relatively wide range of depths from the pelagic zone to the benthic zone, we conducted a comprehensive analysis across both horizontal and vertical gradients. As a result, we detected a diverse array of zooplankton species and observed that certain taxa were dominant in specific regions, reflecting local geographic characteristics. The Ross Sea region, a designated MPA, hosts a unique marine ecosystem that requires ongoing conservation efforts to ensure its resilience against future environmental changes, particularly those driven by climate change. eDNA analysis, with its application in extreme environments like Antarctica, provides an effective, non-invasive method for biodiversity analysis. This approach should be employed for continuous and systematic ecosystem monitoring and management on an international scale. The genetic data of Antarctic species obtained in this study will serve as a foundation for more accurate and efficient analyses of the Antarctic ecosystem in the future.

Supplemental Information

Supplemental Information 1 Results of the environmental parameters measured in the Ross Sea sampling region

Temp: temperature (°C); Sal: salinity (PSU); DO: dissolved oxygen (umol/kg)

Supplemental Information 2 Sequencing and bioinformatic analysis results information of each sample

Supplemental Information 3 Number of OTUs and reads assigned for each taxonomical classification level

The percentage in parentheses indicates the proportions relative to the total number of OTUs and reads, respectively.

Supplemental Information 4 Taxonomic analysis result of zooplankton in order level

The samples of each site were mainly composed of 44 orders of zooplankton.

Additional Information and Declarations

Competing Interests

Author Contributions

Data Availability

The authors declare there are no competing interests.

Soyun Choi performed the experiments, analyzed the data, prepared figures and/or tables, and approved the final draft.

Eunkyung Choi analyzed the data, authored or reviewed drafts of the article, and approved the final draft.

Minjoo Cho performed the experiments, prepared figures and/or tables, and approved the final draft.

Seung Jae Lee analyzed the data, authored or reviewed drafts of the article, and approved the final draft.

Inseo Kim performed the experiments, prepared figures and/or tables, and approved the final draft.

Doyoon Shin performed the experiments, prepared figures and/or tables, and approved the final draft.

Jangyeon Kim performed the experiments, prepared figures and/or tables, and approved the final draft.

Hyoung Sul La conceived and designed the experiments, authored or reviewed drafts of the article, and approved the final draft.

Jae-Sung Rhee conceived and designed the experiments, authored or reviewed drafts of the article, and approved the final draft.

Jeong-Hoon Kim conceived and designed the experiments, authored or reviewed drafts of the article, and approved the final draft.

Hyun Park conceived and designed the experiments, authored or reviewed drafts of the article, and approved the final draft.

The following information was supplied regarding data availability:

Sequence data are available at NCBI, BioProject PRJNA1267705: SRR33694441–SRR33694488.

Raw data is available in the Supplemental files.

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
