# Peer review of "Metabarcoding analysis reveals hidden eukaryotic plankton biodiversity in the Ross Sea, Antarctica"

_PeerJ, doi:10.7717/peerj.20118_

## Round 0.1 · original submission · Major Revisions

The three professional reviewers gave excellent review comments. I believe these comments play a crucial role in thoroughly improving the quality of this manuscript. You need to make revisions one by one in response to these review comments.

Reviewer 1 ·

Basic reporting

Clarity and Language: The manuscript is well-written. The structure conforms to PeerJ standards, and the figures/tables are relevant, high-quality, and well-labeled.
Introduction and Background: The introduction provides a comprehensive context for the study, highlighting the importance of the Ross Sea MPA and the utility of eDNA metabarcoding. The literature is relevant.
Raw Data: The manuscript mentions the availability of raw data. However, the documents provided in the supplementary material (Tables S1-S3) can not be used to reproduce this work.

Experimental design

Research Question: The research question is generally well-defined, relevant, and addresses a clear knowledge gap—the need for comprehensive biodiversity assessment in the Ross Sea MPA using eDNA.
Methods: The methods are described with sufficient detail. The sampling strategy, eDNA extraction, PCR amplification, and bioinformatic analysis are robust and appropriate.
Ethical and Technical Standards: The study adheres to high technical and ethical standards, with non-invasive sampling and rigorous bioinformatic processing.

Validity of the findings

Data: As mentioned above, I can not see the raw data. The statistical analysis is generally robust (e.g., alpha/beta diversity metrics, PCoA, UPGMA). The use of SILVA for taxonomic classification is appropriate.
Novelty and Impact: The study successfully identifies previously undetected species in the Ross Sea, demonstrating the power of eDNA metabarcoding. This contributes significantly to Antarctic biodiversity databases.

Additional comments

My major concern is the manuscript should briefly acknowledge potential limitations, such as primer bias in metabarcoding or the challenges of assigning taxonomy from eDNA data alone.
Meanwhile, Why COI was not selected for eDNA analysis when you targeting zooplankton? L291-301 showed the zooplankton species identified, I am surprised that nearly all these species were rare species in previous traditional morphological analysis. The authors should at least add one paragraph in the Discussion section to mention the shortcoming of the primer used.

I would also suggest the authors to conduct more analysis to show the impact of environmental factors (shown in Table S1) on the plankton.

Specific comments:
Were field blanks and replicates taken at each sampling station?
Figure 1: What does the white line indicate?
Figure 3: Using specific color to show different station groups in the right panel based on the color used in the left panel.
Table 1: use two decimal places to show the longitude and latitude.

Reviewer 2 ·

Basic reporting

Please refer to the review comments for details.

Experimental design

Please refer to the review comments for details.

Validity of the findings

Please refer to the review comments for details.

Additional comments

Please refer to the review comments for details.

Annotated reviews are not available for download in order to protect the identity of reviewers who chose to remain anonymous.

Reviewer 3 ·

Basic reporting

no comment

Experimental design

no comment

Validity of the findings

no comment

Additional comments

This study by Choi et al. revealed eukaryotic plankton diversity in the Ross Sea using 18S rRNA gene amplicon sequencing. This study had special emphasis on the alpha and beta diversity of both phytoplankton and zooplankton. The authors found that phytoplankton communities were influenced by currents and environmental factors, while zooplankton communities displayed horizontal and vertical patterns. My major concerns and detailed comments are shown as below.

Major concerns:
The major concern was the classification of phytoplankton and zooplankton throughout this study. First, I am confused by how to clarify phytoplankton and zooplankton based on metabarcoding data, which was totally missing in the material and methods of this study. Moreover, I strongly felt that the authors were actually talking about ‘pigment’ and ‘non-pigment’ communities. The characteristics of marine eukaryotes are extremely complicate, mainly including autotrophs, mixotrophs and heterotrophs. This study appeared to divide them into phytoplankton and zooplankton roughly, which is not accurate. For example, many groups are rarely classified as zooplankton, such as Syndiniales which can be dominant in many habitats. Second, the primer set here is indeed universal of all eukaryotes (Stoeck et al. 2010, DOI: 10.1111/j.1365-294X.2009.04480.x), which has been widely used. It performed very well especially in protist-based studies. It might be acceptable to analyze phytoplankton and zooplankton communities using data derived from this primer set. However, this primer set may not be a good choice for a study strongly emphasizing on phytoplankton and zooplankton, especially when these two groups were not clearly defined. Therefore, the big limitation in this aspect cannot be ignored and must be reminded in this manuscript. Some additional comments which are also related to the major concern are shown in the minor comments below.

Minor comments:
L1–2: ‘Metagenomic’ is not accurate when amplicon sequencing is actually used. This point has been recently clarified by Bindels et al. (2025, DOI: 10.1186/s40168-025-02091-0).

L40: It is ‘18S rRNA gene’.

L44: The conclusion regarding ‘ocean currents’ was not directly supported, as shown in later contents. Overall, I felt these sentences in Results were hollow. For example, please clarify how ‘strongly influenced’ and what ‘significant differences’.

L49–56: Some bullet points closely related to the findings should be shown, to avoid to sound hollow.

L93: Many ‘research’ throughout this manuscript may be better to be replaced by ‘study/studies’.

L142–144: Please be informed that the sequence of TAReuk454FWD1 was incorrect. Please be informed that the raw origin of this primer set is Stoeck et al. (2010, DOI: 10.1111/j.1365-294X.2009.04480.x).

L144–156: I am strongly confused by the thermal cycling conditions. It seems that a two-step PCR protocol was used. I understand that the second PCR was performed to insert adapter and index sequences (L151), but why so many (30; L155) cycles were used. Normally, 5–10 cycles are enough in this step, and the less, the better. This is because the sequencing data may be biased by too many cycles, considering that a total of 55 cycles were conducted in this study. This is rarely done in previous studies.

L162: As shown in the major concern, how to split the data into phytoplankton and zooplankton groups was totally ignored. Please add the criteria carefully.

L173–174: Similarly, why was this strategy used? This is very confusing. Was this caused by too low counts of ‘phytoplankton’ reads in other samples? If so, I guess this is partly due to data analysis protocol which was based on all eukaryotes. In detail, ‘phytoplankton’ in non-surface samples may suffer from the representativeness of metazoa. Normally, protists and metazoa are individually analyzed in previous studies to avoid this phenomenon. Therefore, a followed concern arose in L175–183. Please clarify the sequence ranges and rarefaction methods for alpha and beta diversity-related analyses. So far, this point was ignored although total counts were shown in Table S2.

L205–206: This is not a good writing style, but repeatedly done in this manuscript (e.g., L240–241, 256–257, and etc.). When mentioning some results, it is not good to write as ‘xxx was shown in Fig. n’. Please rephrase these sentences.

L216–217: Why was the unweighted index (Jaccard) used here, while the weighted index (Bray-Curtis) was used latter (L230)? These contrasting strategies in data interpretations should be carefully clarified. Otherwise, data analyses sound arbitrary.

L222: This represents an important and interesting point in this study. However, there is no direct supports to conclude it. This makes it sounds like a speculation, not a true ‘result’. Please clarify these concluding remarks throughout the manuscript.

L256–257: Were read numbers (not relative abundance) used in Fig. 6? If so, this is rarely done in microbial ecology studies. Please clarify this confusing point.

L266–273: Due to the lack of details of rarefaction, these were difficult to follow.

L282–301: In general, I got lost in this paragraph. For example, what do you mean by saying ‘identified 25 species of phytoplankton and 25 species of zooplankton’? Were these conclusions based on the sequence assignments? If so, this is too vulnerable. Any conclusion down to the species level should be very careful in amplicon-based data (~400 bp here). Please add more information to support these results.

L285–287: This sentence is hard to follow. Throughout this paragraph, it is unclear how to determine ‘previously observed’. Any reference in Table 2?
L287–288: It sounds strange to place this sentence here (compared to the first one in this paragraph).

L291: Considering the unclear definition of ‘zooplankton’ (e.g., including many parasites), it is hard to agree with ‘larger than phytoplankton’. In the same time, ‘zooplankton inhabits relatively deeper waters’ is meaningless here. This vulnerable conclusion was only to support the arbitrary data analysis protocol (i.e., phytoplankton was removed from deeper samples). Please be informed that many studies have expanded the habitat range of marine phytoplankton (Hoppe et al. 2024, DOI: 10.1038/s41467-024-51636-8), including deep oceans (Agusti et al. 2015, DOI: 10.1038/ncomms8608).

L293: What did ‘observed with naked eye’ mean? Did the authors collect microscopic samples? Otherwise, how to draw this conclusion?

L293–295: I cannot follow this sentence, taken many confusions above together.

L301: Any references in Table 2?

L305 and 316: These two paragraphs sounded like ‘Introduction’. Please fundamentally rephrase them.

L347: As concerned above, please show direct and solid supports to draw this conclusion (e.g., water mass analyses) (e.g., Monier et al. 2015, DOI: 10.1038/ismej.2014.197), but not ‘likely’. Otherwise, this should be turned down or removed, and cannot represent one of the main findings at least. It sounds too arbitrary relying on some manually-drawn lines in the map.

L358–360: This sentence is delivering correct information, while moved too far away from the correlations (sounds like overinterpretation).

L376: Yes, they are widespread parasites. Is it proper to define them as ‘zooplankton’?

L385: Correct the insertion of the citation.

L390–391: First, did Arthropoda observed here belong to copepods which are easily detected by this primer set? If so, they are not ‘micro-benthic invertebrates’ but planktons. Again, if the read ranges, rarefaction strategies, and (relative) abundances were not clearly established, these conclusions were vulnerable. For example, this phenomenon easily arose when the number of protist sequences is low in this sample.

L394: Rephrase ‘distinct differences’.

L407: See related comments above, this conclusion sounds arbitrary, because it was strongly affected by your data analysis protocol.

Table 2: References are needed, and the paragraph associated with it should be fundamentally rephrased.

Data availability: Please submit your sequencing data to a public database such as SRA (NCBI).

---

## Round 0.2 · accepted · Accept

Both the reviewers and I have examined your revised manuscript. The new version has addressed all previous concerns and now meets the criteria for acceptance.

Reviewer 1 ·

Basic reporting

This manuscript conforms to PeerJ standards, and the figures/tables are relevant, high-quality, and well-labeled. The literature is relevant. The manuscript mentions the availability of raw data.

Experimental design

The research question is generally well-defined, relevant, and addresses a clear knowledge gap—the need for comprehensive biodiversity assessment in the Ross Sea MPA using eDNA.

Validity of the findings

The statistical analysis is generally robust (e.g., alpha/beta diversity metrics, PCoA, UPGMA).

Reviewer 2 ·

Basic reporting

I have reviewed the revised manuscript and confirm that the authors have satisfactorily addressed all my previous concerns. I recommend ​​acceptance in its current form​​.

Experimental design

I have reviewed the revised manuscript and confirm that the authors have satisfactorily addressed all my previous concerns. I recommend ​​acceptance in its current form​​.

Validity of the findings

I have reviewed the revised manuscript and confirm that the authors have satisfactorily addressed all my previous concerns. I recommend ​​acceptance in its current form​​.

Additional comments

None.

Reviewer 3 ·

Basic reporting

no comment

Experimental design

no comment

Validity of the findings

no comment

Additional comments

My comments have been substantially addressed.